# Probing Memorization of Tabular In-Context Learning

**Francesco Capano** [1]   **Jonas Böhler** [1]

## Abstract

Large tabular models (LTMs), i.e., tabular foundation models leveraging in-context learning (ICL), achieve state-of-the-art performance on tabular tasks. While LLMs are known to unintentionally memorize training data, the memorization dynamics of LTMs remain largely unexplored. We investigate the potential for parametric memorization in tabular ICL. We introduce ICLMEM, a probing framework designed to separate context-based predictions from parametric memorization. Our zero-information multiple-choice context strips away valid contextual patterns to force the model to fall back on its parametric memory. Our controlled fine-tuning setup establishes membership ground truth and accounts for common pitfalls, e.g., distribution shift, feature contamination, base-rate fallacy, and the pre-trained base model acts as reference to calibrate for sample difficulty. Our controlled evaluation on a leading real-world-trained LTM detects moderate memorization signals in 8 out of 10 tasks (AUC up to 0.67 and TPR at $1\%$ FPR $> 0.1$). Notably, memorization signals are strongest for low-cardinality and binary tasks. However, they largely vanish under realistic training conditions. Our findings show LTM memorization signals under specific circumstances (single-task fine-tuning with fixed samples across many epochs and small query size). To protect sensitive data, appropriate measures must be taken, which we discuss.

## 1. Introduction

Tabular data dominate enterprise applications and domains such as healthcare and finance (Chui et al., 2018). Large tabular models (LTMs) (Van Breugel & Van Der Schaar, 2024; Hollmann et al.; Spinaci et al.), i.e., tabular foundation

---

[1]SAP SE, Walldorf, Germany. Correspondence to: Francesco Capano <francesco.capano@sap.com>, Jonas Böhler <jonas.boehler@sap.com>.

*Proceedings of the $2^{nd}$ ICML Workshop on Foundation Models for Structured Data*, Seoul, South Korea. 2026. Copyright 2026 by the author(s).

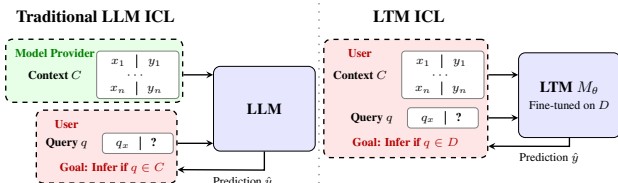

*Figure 1.* Memorization probing for LLM vs LTM. **Left:** In LLMs with ICL the model provider supplies a sensitive context $C$, which the user attempts to extract via user-controlled queries. **Right:** In LTMs, the user controls both the context $C$ and the query $q$. In ICLMEM the context is used to probe whether $q$ was memorized from the sensitive fine-tuning data $D$.

models, achieve state-of-the-art performance for tabular predictions by leveraging in-context learning (ICL) to perform zero-shot predictions on unseen tasks (Brown et al., 2020). At inference, the model receives a context (e.g., historical example rows) alongside a query row with a missing target field to be predicted. While LTMs are typically pre-trained on synthetic data (Hollmann et al.; Qu et al., 2025), recent models incorporate real-world tabular corpora (Spinaci et al.; Garg et al., 2025; Ma et al., 2024), e.g., by fine-tuning on diverse real-world task mixtures to improve performance (Grinsztajn et al., 2025). While LLMs are known to unintentionally memorize training data (Chen et al., 2026; Hayes et al., 2025; Szep et al., 2026), the memorization dynamics of LTMs remain largely unexplored. Rigorous memorization probing is crucial to inform potential mitigations for sensitive fine-tuning data, such as differential privacy and adaptation of data pre-processing and training regimes.

**Problem Setting.** In the ICL paradigm, a model predicts a missing target value $y_q$ given query's input features $x_q$, and a context $C = \{(x_1, y_1), \ldots, (x_n, y_n)\}$ of examples. With LTMs, the user provides both the context $C$ and the query $q$, as shown in Fig. 1. We assume an LTM $M_\Theta$ is fine-tuned on sensitive $D$ by a data owner who provides access to the fine-tuned LTM to users. This setting opens up the possibility to manipulate $C$ arbitrarily, to probe differences in model behavior on context-query pairs derived from data included in or excluded from $D$. In contrast, existing works on ICL privacy primarily assume the model provider defines a sensitive $C$, which an attacker attempts to extract via membership inference attacks (MIA) by manipulating $q$ (Wen et al., 2024; Duan et al., 2024). MIAs (Shokri et al., 2017) aim to infer whether a specific record was part of the training data, and transferring MIAs to

LTMs presents unique challenges. Unlike LLMs, which can generate arbitrary text on a vast output space to extract memorized tokens (Carlini et al., 2021), LTMs are encoder-only architectures that strictly constrain predictions to the pre-defined candidate labels provided in $C$ (Spinaci et al.; Hollmann et al.). More importantly, while masked-language modeling can induce memorization (Hartmann et al., 2023), LTMs operate as Prior-Data Fitted Networks (PFNs) (Müller et al., 2021). They are trained to infer structural connections within the context rather than memorize explicit targets (Müller et al., 2021). This raises a fundamental, yet unanswered question: *What memorization profile do LTMs, inherently trained to perform ICL, have?* To investigate memorization capabilities of LTMs, we present ICLMEM and apply MIA. Specifically, we test membership of a candidate set of query rows by manipulating $C$ as an active probing mechanism (Fig. 1). We hypothesize that by carefully modifying $C$ to eliminate discriminative information, it disrupts the model's context-derived prediction capabilities. Forced to abandon in-context reasoning, the model falls back on its memorized parametric knowledge. For member rows in $D$, we expect predictions to exhibit higher confidence and remain robust to context manipulations (Choquette-Choo et al., 2021; Yeom et al., 2018). We identify specific circumstances for memorization in LTMs (Sec. 4.2) and discuss protection measures (Sec. 6).

## 2. Related Work

While LTMs (Hollmann et al.; Qu et al., 2025; Spinaci et al.; Garg et al., 2025) rapidly advance, investigations into their privacy properties remain limited (Nayyeri et al., 2026). Existing LTM privacy literature primarily targets test-time evasion and adversarial robustness rather than memorization (Simonetto et al., 2024; Djilani et al., 2025; Anwar et al., 2024). Conversely, ICL privacy research largely focuses on LLMs, probing either the leakage of hidden context examples (Wen et al., 2024; Duan et al., 2024) or table memorization within text corpora (German et al., 2025; Bordt et al., 2024). To assess memorization, state-of-the-art MIAs typically rely on shadow models (Carlini et al., 2022a; Zarifzadeh et al., 2024). However, training shadow models for large models is computationally prohibitive (Hayes et al., 2025) and yields unreliable false-positive bounds when the pre-training distribution is unknown (Zhang et al., 2025). Towards privacy assessment of LTMs, we adapt techniques from robustness-assessing MIA (Choquette-Choo et al., 2021) and attribute inference (AIA) (Jayaraman & Evans, 2022; Annamalai et al., 2024; Salem et al., 2023). While AIA aims to infer an unknown sensitive value given a partial record, our goal is inferring membership of a record. To assess memorization in LTMs, we adapt AIA's multiple-choice testing and label randomization and combine zero-information context manipulations with difficulty calibra-

tion via the pre-trained base model (Watson et al., 2022).

## 3. Contributions

We provide the first systematic assessment of memorization in LTMs. Our contributions are:

- We introduce ICLMEM, a probing framework designed to separate context-based predictions from parametric memorization. Our zero-information multiple-choice context strips away contextual patterns forcing the model to fall back on parametric memorization.
- We avoid common MIA false positives by accounting for distribution shifts (via data pre-processing), feature contamination (via data deduplication), base-rate confounding (via label randomization), and intrinsic low-complexity samples (via difficulty calibration).
- We perform an extensive evaluation on LTM ConText-Tab across 1,128 fine-tuning configurations on 10 diverse classification and regression tasks from CARTE.
- We detect moderate memorization in 8 of the 10 tasks (with AUC $> 0.5$ and TPR$_{@r\%} > 0.1$ for $r \in \{1, 10\}$) in our controlled setup with single-task fine-tuning and fixed context-query pairs. The signal is higher for low-cardinality and binary tasks with small training query sizes ($Q = 50$), but largely vanishes under practical setups (e.g., larger $Q = 512$, multi-dataset training, random context-query sampling) (App. D).

Overall, our findings show memorization capabilities in LTMs which help to inform appropriate protection measures. Next, we introduce our methodology and setup (Sec. 4), detail our evaluation (Sec. 5), and conclude with limitations, mitigations, and future works (Sec. 6).

## 4. Methodology and Setup

We introduce ICLMEM, a framework designed to probe memorization in LTMs and mitigate common MIA evaluation pitfalls (Tab. 4, App. A.1). First, we outline unique challenges of assessing LTMs. Then, we discuss the two stages of ICLMEM (Fig. 3, App. A). Namely, controlled fine-tuning with careful data pre-processing and fixed query-context pairs across epochs (Sec. 4.1) and memorization probing with a suite of context manipulations calibrated against the pre-trained base model as reference (Sec. 4.2).

**Challenges in Assessing LTMs.** LTMs rely on ICL to infer structural feature-target correlations from the context, a mechanism that intuitively acts as a barrier to accessing the model's parametric memory. Furthermore, unlike LLMs with massive output vocabularies, LTMs are encoder-only architectures strictly constrained to outputting probabilities over the specific candidate labels present in the context. Our context manipulations address these challenges via zero-information multiple-choice protocol (Sec. 4.2).

### 4.1. Establishing a Memorization Ground Truth

IcLMem pre-processes data to eliminate artifacts, and performs controlled fine-tuning to induce memorization.

**Data Pre-Processing.** To avoid data artifacts being interpreted as memorization signal, IcLMem carefully pre-processes data. First, we deduplicate records to prevent *feature contamination*, i.e., identical records appearing in both member and non-member sets, which inflate the memorization signal. Second, we ensure distributional closeness between disjoint member and non-member splits to preclude *distributional shifts*. Specifically, for target label distributions via total variation distance TVD $< 0.05$ (Gibbs & Su, 2002) for classification and the Kolmogorov–Smirnov statistic KS $< 0.05$ (Massey Jr, 1951) for regression (i.e., difference between empirical cumulative density functions) (Tab. 6, App. A.2). To mitigate the *base-rate fallacy*, i.e., model exploits label distribution to predict likely outcomes, we apply two measures. First, we enforce a uniform label distribution in the training context, preventing the model from minimizing loss by collapsing to majority-class predictions. Second, following (Annamalai et al., 2024), we apply label randomization, i.e., scrambling target labels while preserving marginal distributions, forcing the model to learn statistically improbable mappings.

**Controlled Fine-Tuning.** To induce memorization, we fine-tune on fixed context-query pairs for up to 500 epochs. Specifically, each training step processes a static context set $C'$ with a batch of query rows of size $Q$. We fine-tune across varying learning rates $\eta \in \{10^{-1}, 10^{-2}, 10^{-3}, 10^{-4}\}$ and query sizes $Q \in \{50, 512\}$. We vary $Q$ in training to test its influence on memorization. Smaller $Q$ should amplify the contribution of individual query rows, while larger $Q$ should dilute them. Our evaluation supports this dynamic, showing that larger $Q = 512$ hinders memorization (Sec. 5).

### 4.2. Probing Memorization with IcLMem

The probing stage aims to detect memorization signals by combining zero-information context manipulations with difficulty calibration via the base model.

**Multiple-Choice Protocol.** To separate expected context-based model predictions from unintended parametric memorization, we propose a zero-information strategy mimicking a multiple-choice question. For each query row $q = (x_q, y_q)$ and candidate label set $Y$, we construct a probing context $C_{base}$ containing exactly $|Y|$ copies of the query's features $x_q$, each paired with a distinct candidate label $y_i \in Y$, thereby providing the model with all possible candidate targets. Our hypothesis is that if we strip away all valid contextual patterns, the model will be forced to abandon in-context inference and default to its parametric memory, selecting the target value observed during fine-tuning.

**Context Manipulations.** To assess robustness of predictions across multiple runs, we subject $C_{base}$ to a suite of 48 context manipulations $\Pi$, including positional biasing, distractor injection, and distributional shifts by, e.g., increasing the counts of true target labels, or non-true labels (detailed in App. A.3). We compute the probe loss $L_k(q) = \mathcal{L}(M_\theta(x_q \mid C_k), y_q)$ for each manipulated context $C_k \in \Pi$. The robustness score is defined as the average loss across all manipulations $S(q) = \frac{1}{|\Pi|} \sum_{\pi_k \in \Pi} L_k(q)$, which is the basis for our memorization metric.

**Difficulty Calibration.** An inherently easy query (e.g., with obvious target from its features) has a low loss, regardless of its membership in $D$. To account for this, we follow the *difficulty calibration* approach of Watson et al. (Watson et al., 2022) and compare robustness score from the fine-tuned model, $S(q)$, to that of the pre-trained base model, $S_{ref}(q)$, and compute its delta as $\Delta S(q) = S(q) - S_{ref}(q)$. Notably, we compute $S(q)$ not only using the loss but also using distributional metrics, i.e., prediction entropy and target confidence, which remain robust discriminators even with fine-tuning-induced artifacts (e.g., loss inversion, Sec. 5).

**Memorization Metric.** We compute AUC over the $\Delta S$ scores (denoted AUC($\Delta S$) or simply AUC), and consider AUC $> 0.5$ as the threshold for *detectable* memorization. Unlike high AUC baselines for LLMs (Duan et al., 2024), LTMs operate over a constrained output space. Hence, we adopt this threshold to capture any calibrated signal above chance (Liu et al., 2025). Furthermore, since global AUC averages across all records and can obscure localized memorization (Carlini et al., 2022a), we report the true positive rate at low false positive rates (TPR$_{@r\%}$ for $r \in \{1, 10\}$) providing a granular view of more confident signals.

## 5. Empirical Evaluation

Next, we summarize our results of IcLMem, and defer full details to App. C due to space constraints.

**Setup.** We evaluate IcLMem on ConTextTab (Spinaci et al.) on 10 tasks from CARTE (Kim et al., 2024). We select these tasks as they were not used in pre-training and represent a wide variety of domains and data distributions (details in Tab. 5). We test for memorization across epochs to capture the evolution of the memorization over time resulting in 1128 probing runs (configurations in Tab. 2).

**Isolating Memorization Outcomes.** Our evaluation yields five distinct outcomes (summarized in Tab. 3). Impractical fine-tuning configurations (e.g., large learning rate) can induce data artifacts mimicking memorization. The most prominent is *representational collapse* (affecting ≈55% of runs at $Q = 50$), where aggressive learning rates ($\eta \geq 10^{-2}$) destroy predictive capabilities, yielding near-constant outputs that artificially inflate loss. Difficulty

*Table 1.* ICLMEM detectable memorization configurations (Q=50 except $^\dagger$ with $Q = 512$). Full results in Tabs. 10–11 in App. C.

| Dataset | LR | Ep. | AUC($\Delta S$) | TPR$_{@10\%}$ | TPR$_{@1\%}$ |
|---|---|---|---|---|---|
| buy_buy_baby | $10^{-4}$ | 10 | 0.61 | 0.14 | 0.12 |
| chocolate_bar | $10^{-3}$ | 10 | 0.61 | 0.18 | 0.02 |
| babies_r_us | $10^{-4}$ | 50 | **0.66** | 0.02 | 0.00 |
| beer_ratings$^\dagger$ | $10^{-4}$ | 50 | 0.59 | **0.49** | **0.17** |
| bikedekho$^\dagger$ | $10^{-4}$ | 10 | 0.58 | 0.15 | 0.02 |

calibration identifies these false positives: if $M_{ref}$ achieves identical zero-shot separation, the signal reflects pre-existing dataset bias, not memorization. A secondary artifact is *loss inversion* ($\approx 19\%$ of runs), where pre-trained structural biases drive member loss higher than non-member loss. A detailed breakdown of all outcomes, including context overfitting, is deferred to App. B.2, with results for all datasets in Tabs. 10–11. After accounting for these fine-tuning artifacts, we isolate memorization signal in 12.0% of runs at $Q = 50$ (Tab. 5), and 8 of the 10 evaluated tasks in total under typical fine-tuning configurations ($\eta \leq 10^{-3}$), exemplified in Tab. 1. More granularly, the TPR at low FPR metric reveals localized memorization. On beer_ratings at $Q = 512$, ICLMEM achieves only AUC $= 0.59$ but identifies 49% of member records at 10% FPR; this dataset retains detectable memorization at $Q = 512$ due to its high target cardinality and small size, as analyzed below. On buy_buy_baby, we reach TPR$_{@1\%} = 0.12$, indicating that in some instances fine-tuning records can be confidently distinguished.

**Drivers of Memorization.** We find that memorization depends on the interaction between query size $Q$, target cardinality $k$, and dataset size $n$. Firstly, expanding $Q$ mitigates memorization in our evaluation: at $Q = 512$, the overall detectable rate drops to 1.8% (a $6.7\times$ reduction versus $Q = 50$), as larger queries stabilize batch variance and enforce stronger zero-shot inductive biases. Secondly, at low query sizes ($Q = 50$), memorization is driven by *label density* ($n/k$, the average records per class). Binary tasks ($k = 2$, e.g., chocolate_bar_ratings) concentrate the gradient signal for specific classes per step, yielding the highest vulnerability (up to 55% detectable runs). As $k$ grows, this concentration dissolves. Finally, high-cardinality tasks require two conditions to trigger memorization: high *query coverage* ($Q/n$, the dataset fraction seen per step) and a target that is *not structurally deducible*. If a target cannot be logically inferred from its features alone (e.g., exact prices or subjective float ratings in beer_ratings), the model is forced to fall back on parametric memory, provided $Q$ covers enough of the dataset. Conversely, if a target can be deduced via ICL (e.g., text-statistic formulas in clear_corpus), the model relies on in-context reasoning and yields no memorization signal, regardless of $Q$.

**Temporal Evolution of Memorization.** Across fine-tuning epochs (Fig. 2), the $Q = 50$ signal emerges at epoch 1 (4.2% of configurations) and grows to 20% by epoch 250. At $Q = 512$, memorization is largely suppressed: it peaks

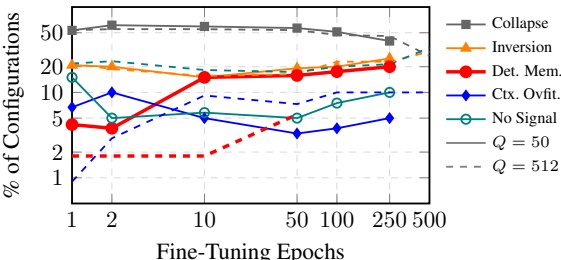

*Figure 2.* Temporal evolution of ICLMEM outcomes

at 5.5% around epoch 50 and vanishes after, showing that larger queries reduce memorization and accelerate its decay.

## 6. Discussion and Conclusion

Next, we discuss limitations, mitigations, and future work.

**Limitations.** First, while ICLMEM detects memorization, the signal is moderate: only 20% of tasks show TPR$_{@1\%} > 0.1$. Second, our probe uses an artificial setup (single task, fixed context-query pairs) to induce memorization. In contrast, realistic LTM regimes pre-train on large task mixtures for only 2–5 epochs (Spinaci et al.), which dilutes memorization. We empirically verify that under joint multi-task training with random context-query sampling at $Q = 512$, detectable tasks drop from 5 to 1, with the signal appearing only at epoch 10, i.e., beyond realistic budgets (App. D). Finally, while our evaluation is broad, our scope is currently limited (1 model, 10 tasks) and to be expanded.

**Mitigations.** To protect sensitive data, appropriate measures must be taken. Differential privacy (e.g., DP-SGD (Abadi et al., 2016)) provides rigorous privacy guarantees that requires careful privacy-utility trade-offs but also shows promising utility when fine-tuning a public base model (Yu et al., 2021; Li et al., 2021). Additionally, our findings show LTM memorization in certain training conditions that can be avoided (e.g., small query size with fixed query-context pairs over many epochs on single task, App. D) or inform data filtering of memorization-prone samples (albeit with potential for privacy onion effects (Carlini et al., 2022b)).

**Conclusion and Future Work.** We initiated the systematic study of memorization in LTMs. In our controlled setup ICLMEM detects moderate memorization signal in 8 out of 10 evaluated tasks (AUC up to 0.67 and TPR at 1% FPR $> 0.1$). Our results demonstrate that, under specific conditions, tabular foundation models can memorize data, underscoring the need for appropriate protective measures when sensitive data is involved. In future work, we aim to expand the evaluation to other LTMs (Qu et al., 2025; Hollmann et al.) and datasets (Klein et al.). We also aim to design *data canaries* (i.e., crafted samples with unique fingerprints or improbable feature-label mappings (Carlini et al., 2019)) to amplify memorization signal and bound the worst-case detection limits of ICLMEM.

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

## A. Detailed Probing Methodology and Pipeline

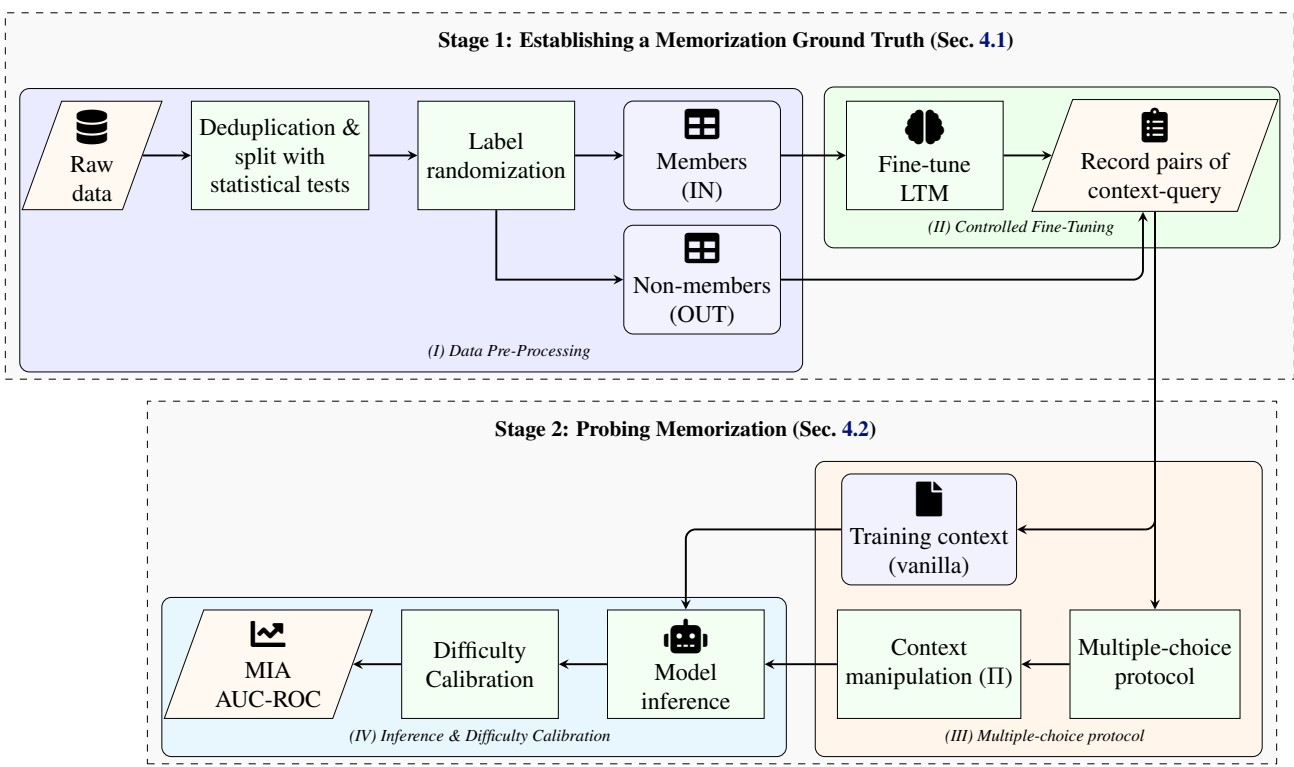

*Figure 3.* End-to-end ICLMEM pipeline. **Stage 1**: (I) *Data pre-processing* — deduplication, statistical split validation, member/non-member assignment, and optional label randomization as a null-hypothesis control (applicable to members, non-members, or both); (II) *Controlled fine-tuning* — LTM fine-tuning on static context-query pairs and logging of all context-query pairs for verified ground truth. **Stage 2**: (III) *Multiple-choice protocol* — zero-information context construction and adversarial context manipulation suite Π; (IV) *Inference & difficulty calibration* — model inference under both the training (vanilla) context and Π, difficulty calibration with instance-level delta score $\Delta S(q) = S(q) - S_{ref}(q)$ computation against the pre-trained reference model $M_{ref}$.

Figure 3 illustrates the ICLMEM pipeline, which consists of two main stages. (I) Establishing a memorization ground truth via data pre-processing and controlled fine-tuning. Here, we create a verified member/non-member split, optionally randomize labels as a null-hypothesis control, and log all context-query pairs for ground-truth verification. (II) Probing memorization using a multiple-choice protocol with adversarial context manipulation. Here, we build a zero-information context, apply a suite of 48 context manipulations, perform model inference under both the vanilla and manipulated contexts, aggregate scores, and compute delta scores against a pre-trained reference model for difficulty calibration. Tab. 2 lists all fine-tuning and probing hyperparameters used in our evaluation. Tab. 3 defines the classification criteria for the five evaluation outcomes.

### A.1. Addressing Common Pitfalls

Tab. 4 summarizes the four common MIA evaluation pitfalls identified by (Evertz et al., 2025) and the corresponding countermeasures implemented in ICLMEM.

### A.2. Dataset Details

Table 5 lists the 10 tasks evaluated in our experiments, all drawn from the CARTE benchmark (Kim et al., 2024). We selected these datasets to cover a heterogeneous mix of real-world classification and regression domains. Crucially, all evaluated datasets are strictly excluded from the ConTextTab pre-training corpus to ensure a valid zero-shot baseline.

**Member/Non-Member Split Validation.** Tab. 6 reports distributional similarity between the member and non-member splits for all 10 datasets at both $Q = 50$ and $Q = 512$. For regression, we calculate the Kolmogorov-Smirnov (KS) statistic (Massey Jr, 1951) on the continuous target values, i.e., the maximum difference between the empirical cumulative distribution

*Table 2.* Experimental hyperparameters for LTM fine-tuning and the ICLMEM protocol.

| Stage | Parameter | Value(s) |
|---|---|---|
| **Fine-Tuning** | Base Model | `ConTextTab` |
| | Optimizer | AdamW |
| | Learning Rates ($\eta$) | $\{10^{-1}, 10^{-2}, 10^{-3}, 10^{-4}\}$ |
| | Maximum Epochs | 500 |
| | Max context size | 512 rows |
| | Context label distribution | Uniform |
| | Query size ($Q$) | $\{50, 512\}$ rows |
| | Query label distribution | Source distribution |
| | Member label randomization | {False, True} |
| **Probing** | Reference model ($M_{ref}$) | Pre-trained `ConTextTab` |
| | Epochs used for probing | {0, 1, 2, 5, 10, 50, 100, 250, 500} |
| | Context size for probing | Depends on the number of unique labels and adversarial manipulations |
| | Query size for probing | 1 query row replicated 50 times |
| | Adversarial manipulations ($|\Pi|$) | 48 distinct configurations (see Tab. 7) |
| | Label randomization | {False, True, Only Members} |
| | Evaluation metrics | Cross-entropy loss, entropy, confidence, $R^2$ |
| | Primary summary metric | AUC($\Delta S$), where $\Delta S(q) = S(q) - S_{ref}(q)$ |

*Table 3.* Classification criteria for the five evaluation outcome categories, applied in priority order (top to bottom). AUC($\Delta S$) is computed on per-instance delta scores $\Delta S(q) = S(q) - S_{ref}(q)$; $\mathcal{L}_v(M, q)$ denotes the *vanilla loss* (loss on the unmanipulated context, without any $\pi_k$); $\mu_{\text{gap}}$ and $\sigma_{\text{gap}}$ are the mean and standard deviation of the per-manipulation probe loss gap $g_k = \bar{\ell}_{\text{atk}}^M(M_\theta, \pi_k) - \bar{\ell}_{\text{atk}}^{NM}(M_\theta, \pi_k)$ across all $\pi_k \in \Pi$.

| Outcome | Key Condition | Intuition |
|---|---|---|
| Representational Collapse | $\mathcal{L}_v(M_\theta) > 5 \times \mathcal{L}_v(M_{ref})$, or accuracy collapses | Fine-tuning destroys predictive ability; the model outputs near-constant predictions. Any separation is a numerical artifact, not memorization. Signal persists unchanged when labels are shuffled. |
| Loss Inversion | $\bar{\ell}_{\text{atk}}^{NM}(M_\theta) < \bar{\ell}_{\text{atk}}^M(M_\theta)$ | A pre-trained structural bias causes non-members to have lower mean probe loss than members across all context manipulations. Loss-based MIA signals are unreliable; distributional metrics (e.g., entropy) should be checked instead. |
| Detectable Memorization | AUC($\Delta S$) $> 0.5$ on loss or entropy | The fine-tuned model exposes a calibrated membership signal above the pre-trained baseline. The signal weakens materially under shuffled labels, confirming it tracks actual label content. |
| Context Overfitting | $|\mu_{\text{gap}}| < 0.05$ and $\sigma_{\text{gap}} < 0.2$ | Fine-tuning encodes dataset schema or label distribution, not individual records. Every context manipulation yields the same near-zero member–non-member probe loss gap (stable null). |
| No Signal | Catch-all (none of the above) | AUC($\Delta S$) $\leq 0.5$ on all metrics with high variance across manipulations ($\sigma_{\text{gap}}$ large). The model does not systematically distinguish members from non-members under any context manipulation. |

*Table 4.* Common MIA evaluation pitfalls from (Evertz et al., 2025; Zhang et al., 2025) and IcLMEM corresponding countermeasures.

| Pitfall | Issue | Countermeasure in IcLMEM |
|---|---|---|
| Base-rate fallacy | Majority-class prediction is mistaken for a membership signal. | Label randomization forces the model to learn improbable mappings; detectable memorization is the only way to predict the randomized label. |
| Distributional shift | Train/test distributions differ, making the attack detect the shift rather than memorization. | Statistical split validation via TVD (classification) and the KS test (regression) ensures members and non-members are identically distributed. |
| Feature contamination | Duplicate rows inflate the attack score by appearing in both member and non-member sets. | Strict deduplication is enforced before partitioning. |
| No calibration | Low sample complexity (easy queries) is confused with memorization. | Adapting the difficulty calibration idea of (Watson et al., 2022) to our shadow-model-free setting, we compute per-instance delta scores $\Delta S(q) = S(q) - S_{ref}(q)$ using the pre-trained reference model $M_{ref}$; only $\text{AUC}(\Delta S) > 0.5$ is treated as a detectable signal. |
| Unverified ground truth | The exact set of records processed during training is unknown, making it impossible to bound the FPR without shadow models (Zhang et al., 2025). | We log every context-query index pair processed during fine-tuning. Only rows provably seen by the model are labeled as members, eliminating label noise from random subsampling or on-the-fly data filtering. |

*Table 5.* Evaluated datasets with target column and task type, i.e., C(lassification) and R(egression), from CARTE (Kim et al., 2024)

| Table Name | Target | Type |
|---|---|---|
| anime_planet | Rating_Score | R |
| babies_r_us | price | R |
| beer_ratings | review_overall | R |
| bikedekho | price | R |
| bikewale | price | R |
| buy_buy_baby | price | R |
| cardekho | price | R |
| chocolate_bar_ratings | Rating | C |
| clear_corpus | BT_Easiness | R |
| coffee_ratings | rating | C |

functions, to bound the distance between splits. For classification, we compute the Total Variation Distance (TVD) (Gibbs & Su, 2002) over the label distribution. All KS statistics are below $0.05$ (maximum: $0.035$ for buy_buy_baby at $Q = 512$). Furthermore, all KS $p$-values are above the $0.05$ significance threshold (minimum: $0.530$ for buy_buy_baby at $Q = 512$), meaning we fail to reject the null hypothesis that the member and non-member splits are drawn from the same distribution. This confirms that no statistically significant distributional shift exists between our data partitions. Similarly, TVD values for classification tasks are small ($\leq 0.056$), indicating near-identical label distributions across splits. We additionally report the Wasserstein distance for completeness (Marek et al., 2025); note that this metric is scale-dependent and therefore not directly comparable across datasets.

### A.3. Context Manipulations

The probing stage subjects each multiple-choice context to a structured suite of 48 manipulations $\Pi$, designed to degrade context utility and expose any reliance on parametric memory. Tab. 7 enumerates all configurations across 11 categories of manipulation types. Below we describe the 4 main categories.

1. **Positional Shuffling:** Alters the row index of the true target within the context (random, first, or last position). LTMs are designed to be permutation-invariant to row order; a model relying solely on ICL should produce identical predictions regardless of ordering. However, fine-tuning on static, fixed-order context-query pairs can introduce positional biases if optimization exploits ordering shortcuts. This manipulation tests whether any membership signal is feature-encoded in the model's weights or an artifact of positional over-indexing.

*Table 6.* Member vs. non-member distributional similarity per dataset and query size. Regression tasks use a two-sample KS test; classification tasks use TVD over the label distribution. All KS stat values are $< 0.05$ and all KS $p$-values are $> 0.05$, confirming no statistically significant distributional shift between member and non-member splits. TVD values for classification tasks are similarly small ($\leq 0.056$), indicating near-identical label distributions across splits. We additionally report the Wasserstein distance for completeness, but note that this metric is scale-dependent and not directly comparable across datasets with different target ranges.

| Dataset | Task | $Q$ | KS stat | KS $p$ | W | TVD |
|---|---|---|---|---|---|---|
| $Q = 512$ | | | | | | |
| anime_planet | regr. | 512 | 0.029 | 0.611 | 0.025 | — |
| babies_r_us | regr. | 512 | 0.021 | 0.968 | 0.030 | — |
| beer_ratings | regr. | 512 | 0.021 | 0.975 | 0.021 | — |
| bikedekho | regr. | 512 | 0.025 | 0.889 | 0.014 | — |
| bikewale | regr. | 512 | 0.033 | 0.595 | 0.011 | — |
| buy_buy_baby | regr. | 512 | 0.035 | 0.530 | 0.065 | — |
| cardekho | regr. | 512 | 0.028 | 0.763 | 0.007 | — |
| clear_corpus | regr. | 512 | 0.023 | 0.941 | 0.032 | — |
| chocolate_bar_ratings | class. | 512 | — | — | — | 0.044 |
| coffee_ratings | class. | 512 | — | — | — | 0.056 |
| $Q = 50$ | | | | | | |
| anime_planet | regr. | 50 | 0.026 | 0.983 | 0.024 | — |
| babies_r_us | regr. | 50 | 0.020 | 1.000 | 0.023 | — |
| beer_ratings | regr. | 50 | 0.025 | 0.993 | 0.016 | — |
| bikedekho | regr. | 50 | 0.027 | 0.983 | 0.012 | — |
| bikewale | regr. | 50 | 0.044 | 0.622 | 0.013 | — |
| buy_buy_baby | regr. | 50 | 0.027 | 0.984 | 0.041 | — |
| cardekho | regr. | 50 | 0.032 | 0.921 | 0.010 | — |
| clear_corpus | regr. | 50 | 0.027 | 0.988 | 0.038 | — |
| chocolate_bar_ratings | class. | 50 | — | — | — | 0.005 |
| coffee_ratings | class. | 50 | — | — | — | 0.043 |

2. **Distribution Skewing:** Skews the marginal label distribution in the context. *Skew true* over-represents the true target (amplification factor $\in \{2, 3, 5\}$); *Skew incorrect* over-represents $k \in \{1, 2, 3\}$ incorrect targets (factor $\in \{2, 3, 5\}$); *Mixed skewing* adjusts both simultaneously. *Uniform and random scaling* amplify all candidate frequencies uniformly or randomly (factor $\in \{2, 5, 10\}$).

3. **Feature Alteration:** *Feature reduction* drops 2 valid feature columns to obscure pattern recognition; *Distractor injection* appends 1–2 entirely random columns to introduce spurious signal. Both test whether the memorization signal survives changes to feature space.

4. **Combined Stress Tests:** 2-way combinations pair positional shuffling with a single distributional or structural manipulation. 3-way stress tests simultaneously apply spatial, distributional, and structural degradation.

*Table 7.* The complete suite of 48 adversarial context manipulations utilized to evaluate predictive robustness.

| Perturbation Category | Varying Parameters | Count | Description |
|---|---|---|---|
| **Baseline** | N/A | 1 | Unperturbed, zero-information multiple-choice context. |
| **Positional Shuffling** | Position $\in \{\text{Random}, \text{First}, \text{Last}\}$ | 3 | Alters the spatial index of the true target row within the context sequence. |
| **Feature Reduction** | Columns removed $= 2$ | 1 | Drops valid feature columns to obscure pattern recognition. |
| **Distractor Injection** | Noisy columns $\in \{1, 2\}$ | 2 | Appends entirely random/noisy features to the context. |
| **Uniform Scaling** | Replication factor $\in \{2, 5, 10\}$ | 3 | Uniformly amplifies the frequency of all candidate labels in the context. |
| **Random Scaling** | Replication factor $\in \{2, 5, 10\}$ | 3 | Randomly alters the distribution frequencies of all candidate labels in the context. |
| **Skew True Target** | Amplification factor $\in \{2, 3, 5\}$ | 3 | Over-represents the true target label. |
| **Skew Incorrect Targets** | Targets skewed $k \in \{1, 2, 3\}$ Factor $\in \{2, 3, 5\}$ | 9 | Over-represents $k$ incorrect candidate labels. |
| **Mixed Skewing** | True factor $\in \{2, 3, 5\}$ Incorrect factor $\in \{2, 3, 5\}$ | 7 | Simultaneously skews both the true target and incorrect targets. |
| **2-Way Combinations** | Shuffle + {Uniform, Skewed, Random, Reduce} | 9 | Combines positional shuffling with a single distributional or structural manipulation. |
| **3-Way Stress Tests** | Shuffle + Skew/Uniform/Random+ Feature Reduction | 7 | Stress tests simultaneously applying spatial, distributional, and structural degradation. |
| **Total Configurations** | | **48** | |

The count column in Tab. 7 specifies the number of distinct parameter configurations per category; the combinatorial categories (2-way, 3-way combinations) compose multiple single-axis manipulations to stress-test signal robustness simul-

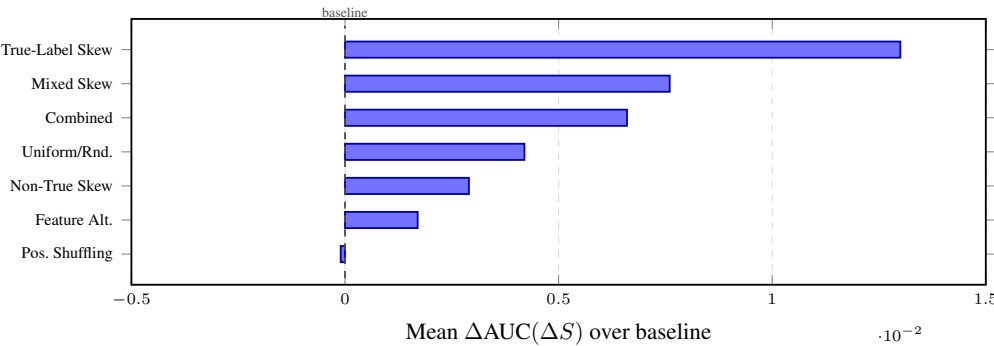

*Figure 4.* Mean $\Delta\text{AUC}(\Delta S)$ relative to the unperturbed baseline ($\text{AUC}_{\text{baseline}}$=0.543), per perturbation category averaged over detectable memorization configurations. True-label distribution skewing provides the strongest average lift ($\Delta$+0.013; best single perturbation: `mixed_true_5_non_true_5`, $\Delta$+0.017). Positional shuffling is indistinguishable from baseline ($\Delta\approx$0).

taneously. Non-members rely entirely on the context for generalization and should exhibit clear loss spikes under strong degradation. Members, whose target mapping is encoded in the model's weights, should maintain a stable probe loss across all manipulations.

**Perturbation Effectiveness.** We evaluate perturbation effectiveness on the detectable memorization configurations, measuring the mean loss-based $\text{AUC}(\Delta S)$ per perturbation. *Distribution skewing* is the most effective single perturbation: amplifying the true target label by $2\times$ yields $\text{AUC} = 0.557$ ($\Delta$+0.015), and simultaneously amplifying all labels by $5\times$ yields the highest observed AUC of $0.559$ ($\Delta$+0.017). This aligns with the design rationale: a memorized member resists context pressure because the correct label is encoded in its weights, whereas a non-member simply follows the over-represented label. Also, as expected, *Positional shuffling* is entirely ineffective ($\text{AUC} \approx 0.543$, $\Delta \approx 0$), confirming that memorization is stored as weight-level feature-label associations rather than positional heuristics, and that the model is indeed permutation-invariant to row order. Feature reduction and distractor injection produce negligible gains (AUC 0.543–0.546). Combining positional shuffling with distribution skewing (`shuffle_and_skewed_2`: $\text{AUC} = 0.557$) does not improve over skewing alone, and heavily over-representing incorrect labels ($5\times$) slightly suppresses the signal (AUC $= 0.539$), likely by reducing prediction confidence across all labels including the true one. No individual perturbation amplifies the signal by more than $\Delta\text{AUC} \approx 0.017$; the practical value of the full perturbation suite lies in aggregating these stable but moderate individual signals into a more robust membership discriminator. Future work could explore more aggressive perturbations (e.g., based on label cardinality or feature importance) to further amplify signal.

### A.4. The Necessity of Difficulty Calibration

Raw, absolute probing metrics are unreliable for auditing memorization (Watson et al., 2022). Pre-trained models already encode strong semantic priors and structural logic, achieving high prediction confidence on inherently easy datasets without task-specific fine-tuning. Relying on absolute metrics confounds this pre-existing structural bias with training-induced memorization.

**Impact on True Positive Rates (TPR).** Across our 530 structurally healthy (non-collapsed) configurations, raw loss evaluation flagged 241 runs with uncalibrated $\text{TPR}_{@10\%} \geq 0.10$. Difficulty calibration against $M_{ref}$ revealed that 230 of these cases (95%) were purely data artifacts: the pre-trained model naturally extracted these records prior to fine-tuning due to inherent sample easiness. Applying the calibrated margin ($\Delta\text{TPR}_{@10\%} \geq 0.10$) reduces this to 11 configurations with detectable memorization.

**Impact on AUC-ROC.** Global distributional metrics exhibit the same bias. If a dataset's features correlate strongly with a target label under the model's pre-training priors, $M_{ref}$ may yield a raw AUC of 0.85 through zero-shot inference alone. An uncalibrated audit would flag a subsequent fine-tuned AUC of 0.87 as a large signal. Computing delta scores $\Delta S(q) = S(q) - S_{ref}(q)$ and then $\text{AUC}(\Delta S)$ correctly identifies this as a structural artifact: since the fine-tuning contributes minimal additional discriminative power, $\text{AUC}(\Delta S) \approx 0.5$, indistinguishable from random guessing. Instance-level calibration via $\Delta S(q)$ is therefore necessary to isolate parametric memorization.

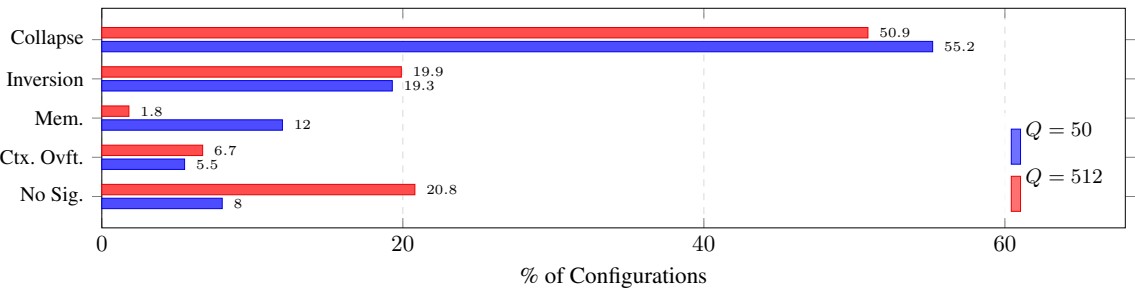

*Figure 5.* Distribution of ICLMEM outcomes. Abbreviations are: *Mem*(orization), *Ctx. Ovft.*: context overfitting; *No Sig*(nal)

# B. Extended Evaluation Analysis

While Section 5 in the main text summarizes the five observed outcomes and their aggregate rates, this appendix provides a deeper analytical breakdown. We first detail the dominant artifact outcomes (representational collapse and loss inversion), which account for the majority of configurations, then analyze the secondary edge-case outcomes (context overfitting and zero-signal runs), track the temporal evolution of the memorization signal across fine-tuning epochs, and explore the disconnect between global and localized memorization.

## B.1. Artifact Outcomes: Collapse and Inversion

**Outcome I: Representational Collapse.** Affecting over half of our configurations ($55.2\%$ at $Q = 50$; $50.9\%$ at $Q = 512$), *representational collapse* is the dominant fine-tuning artifact. At aggressive learning rates ($\eta \geq 10^{-2}$), LTMs lose their in-context reasoning capability and output near-constant predictions. While this yields high uncalibrated AUCs ($> 0.7$), difficulty calibration confirms these are pre-existing dataset biases rather than induced memorization. We formally flag collapsed models using an empirically derived threshold: $\mathcal{L}_{FT} > 5 \times \mathcal{L}_{Base}$. Across our 1,128 runs, the loss ratio ($\mathcal{L}_{FT}/\mathcal{L}_{Base}$) follows a strict bimodal distribution. Stable configurations ($\eta \leq 10^{-4}$) tightly cluster between $1.0$–$2.0\times$, whereas aggressive rates cause rapid degradation, pushing median ratios above $8.0\times$ (and frequently exceeding $10^4$). This $5\times$ boundary robustly separates structurally broken models from true parametric memorization.

**Outcome II: Loss Inversion.** In $19.3\%$ of configurations at $Q = 50$, there is a further artifact: *loss inversion*, where non-members yield lower mean probe loss than members after fine-tuning. This occurs when training loss is already inverted in the pre-trained model, or when the collapsed model's static predictions align better with non-members. Our framework addresses this structural bias by evaluating distributional metrics (i.e., entropy and confidence) which remain discriminative even when loss is inverted.

## B.2. Secondary Dynamics and Edge Cases

Our methodology isolates two additional outcomes beyond detectable memorization and representational collapse.

**Outcome IV: Context Overfitting.** In $5.5\%$ of runs at $Q = 50$, the model learns statistical properties of the dataset schema rather than individual record membership. To classify this outcome, we define $g_k = \bar{\ell}_{atk}^M(M_\theta, \pi_k) - \bar{\ell}_{atk}^{NM}(M_\theta, \pi_k)$ as the mean member minus non-member probe loss gap under perturbation $\pi_k$, and let $\mu_{gap}$ and $\sigma_{gap}$ denote its mean and standard deviation across all 48 manipulations in $\Pi$. The mean probe loss gap ($\mu_{gap}$) is near zero, but $\sigma_{gap}$ is minimal across all 48 context manipulations: the zero is stable on every perturbation, not noisy. This indicates the model treats members and non-members identically under all context manipulations, having overfitted to the table format rather than individual rows.

**Outcome V: No Signal.** This outcome captures configurations ($8.0\%$ at $Q = 50$; $20.8\%$ at $Q = 512$) where ICLMEM produces no systematic pattern. Noisy and inconsistent per-manipulation probe loss gaps ($g_k$) indicate that fine-tuning improved task performance without inducing a detectable row-level membership shift.

## B.3. Overall memorization (AUC) vs confident instance signal (TPR at low FPR)

Our calibrated audit detected memorization (AUC($\Delta S$) $> 0.5$) in 8 of the 10 evaluated tasks under specific hyperparameter regimes, despite representational collapse and chance predictions being the predominant outcomes. Tab. 8 reports the peak vulnerability configuration for each of these 8 tasks: learning rate, epoch, context size ($Q$), and the label randomization condition where the peak signal was detected. Normal indicates the true-label condition; Shuffled (M) indicates member-only

*Table 8.* The peak detectability configurations for the 8 CARTE datasets, i.e., showing detectable memorization. The AUC and TPR metrics are computed directly on delta scores ($\Delta S$), where an AUC of 0.50 represents random guessing.

| Dataset | Label Condition | LR | Epoch | Q | Primary Metric | AUC | TPR$_{@10\%}$ | TPR$_{@1\%}$ |
|---|---|---|---|---|---|---|---|---|
| babies_r_us | Normal | $10^{-3}$ | 1 | 512 | Loss | 0.668 | 0.205 | 0.008 |
| bikewale | Shuffled (M) | $10^{-4}$ | 250 | 50 | Loss | 0.624 | 0.000 | 0.000 |
| chocolate_bar_ratings | Shuffled (M) | $10^{-2}$ | 100 | 50 | Loss | 0.611 | 0.180 | 0.020 |
| buy_buy_baby | Normal | $10^{-4}$ | 10 | 50 | Loss | 0.608 | 0.140 | 0.120 |
| cardekho | Normal | $10^{-4}$ | 1 | 50 | Loss | 0.596 | 0.140 | 0.000 |
| beer_ratings | Shuffled (M) | $10^{-4}$ | 50 | 512 | Loss | 0.586 | 0.494 | 0.166 |
| bikedekho | Normal | $10^{-4}$ | 10 | 512 | Loss | 0.579 | 0.148 | 0.016 |
| coffee_ratings | Normal | $10^{-3}$ | 10 | 50 | Loss | 0.560 | 0.200 | 0.000 |

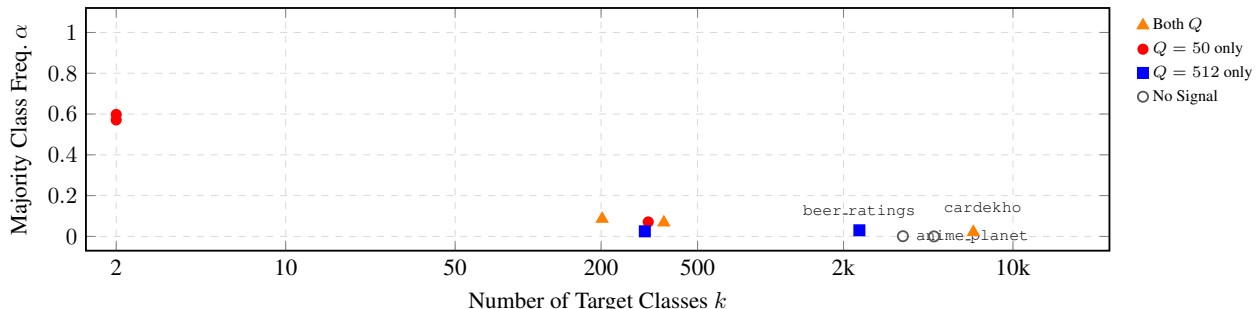

*Figure 6.* Memorization signal category for each of the 10 evaluated CARTE datasets as a function of target cardinality $k$ and majority class frequency $\alpha$. The two points at $k=2$ represent chocolate_bar_ratings and coffee_ratings; the two points near $k=300$ represent bikedekho (Q=512 only, $k = 303$) and buy_buy_baby (Q=50 only, $k = 313$).

label scrambling; Shuffled (All) indicates full-dataset scrambling. We report the calibrated AUC($\Delta S$) to capture global memorization trends across all records, complemented by TPR at low FPR (TPR$_{@10\%}$ and TPR$_{@1\%}$) to measure localized signal of individual samples. We report $AUC$ and $TPR$ for the metric (loss or entropy) yielding the maximum signal.

**Global vs. Localized Memorization Signal.** Several configurations in Tab. 8 yield a positive global memorization signal (AUC($\Delta S$) > 0.5) while simultaneously exhibiting low TPR at strict FPR thresholds. Fine-tuning can induce a global distributional shift that separates members from non-members on average (AUC) without concentrating that signal at the individual-record level: $M_{ref}$ may achieve higher extreme-tail confidence on a subset of records purely through zero-shot structural bias. A detectable global signal therefore does not imply a strong localized one; row-level memorization is confined to a smaller subset of configurations under specific hyperparameter regimes.

### B.4. Data Characteristic Analysis

Fig. 6 plots all 10 evaluated datasets in the target-cardinality–majority-class-frequency space, colored by memorization outcome category. Tab. 9 reports the underlying statistics and per-$Q$ detectable memorization run counts.

**Feature Composition.** CARTE datasets are mixed with both categorical and numerical features: beer_ratings has 14 of 19 features as numeric, i.e., the highest ratio among all memorized datasets, producing near-unique feature vectors that ICL cannot generalize from, contributing to its parametric memorization at $Q = 512$. In contrast, clear_corpus (15/26 numeric) has similar richness but its continuous readability targets are feature-predictable, suppressing memorization. For predominantly categorical datasets, feature type does not discriminate memorized from non-memorized tasks; target cardinality and class concentration remain the primary factors.

## C. Exhaustive Hyperparameter Results

Tabs. 10 and 11 report outcome metrics for $Q = 50$ and $Q = 512$ respectively, covering all 8 CARTE datasets and all evaluated learning rates. For each query size, we report the epoch yielding the highest signal of the reported outcome per learning rate $\eta$, selected independently for each $Q$.

*Table 9.* Dataset statistics and detectable memorization run counts for all 10 evaluated CARTE tasks, sorted by target cardinality $k$. $n$: dataset size; $\alpha$: majority class frequency; Conf@50 / Conf@512: number of configurations yielding detectable memorization at $Q=50$ and $Q=512$, respectively.

| Dataset | $n$ | $k$ | $\alpha$ | Conf@50 | Conf@512 |
|---|---|---|---|---|---|
| chocolate_bar | 2.6k | 2 | 0.571 | 31 | 0 |
| coffee_ratings | 2.1k | 2 | 0.598 | 1 | 0 |
| babies_r_us | 5.1k | 202 | 0.086 | 1 | 3 |
| bikedekho | 4.8k | 303 | 0.025 | 0 | 2 |
| buy_buy_baby | 10.7k | 313 | 0.071 | 19 | 0 |
| bikewale | 9.0k | 363 | 0.068 | 14 | 1 |
| beer_ratings | 3.2k | 2325 | 0.030 | 0 | 3 |
| anime_planet | 14.4k | 3516 | 0.001 | 0 | 0 |
| clear_corpus | 4.7k | 4724 | 0.000 | 0 | 0 |
| cardekho | 37.8k | 6865 | 0.021 | 1 | 1 |

*Table 10.* Exhaustive evaluation of 8 CARTE datasets under ICLMEM with $Q = 50$ context queries. For each dataset, we track metrics across all four learning rates, selecting the peak-signal epoch independently for $Q = 50$. $\Delta\mathcal{L}_v$: vanilla loss delta $\mathcal{L}_v(M_\theta) - \mathcal{L}_v(M_{ref})$ for members (IN) and non-members (OUT). **N**: normal label condition; **S**: member-shuffled condition. AUC($\Delta S$) and TPR are computed on instance-level delta scores $\Delta S(q) = S(q) - S_{ref}(q)$; bold AUC = detectable memorization (AUC($\Delta S$) > 0.5). **Pert.**: mean $\mu_{gap}$ and std. $\sigma_{gap}$ of the per-manipulation member–non-member probe loss gap $g_k$ across all $\pi_k \in \Pi$. **Outcome**: **Mem.** = Detectable Memorization, Coll. = Collapse, Inv. = Inversion, Ctx.Ov. = Context Overfitting, No Sig. = No Signal.

| Dataset | LR | Ep. | $\Delta\mathcal{L}_v$ IN | $\Delta\mathcal{L}_v$ OUT | AUC($\Delta S$) N | AUC($\Delta S$) S | Pert. $\mu_{gap}$ | Pert. $\sigma_{gap}$ | TPR @10% | TPR @1% | Outcome |
|---|---|---|---|---|---|---|---|---|---|---|---|
| babies_r_us | $10^{-1}$ | 10 | $1.9\cdot10^7$ | $1.4\cdot10^7$ | 1.000 | 1.000 | $-3.7\cdot10^5$ | $9.2\cdot10^4$ | 1.00 | 1.00 | Coll. |
| | $10^{-2}$ | 50 | $9.1\cdot10^4$ | $6.8\cdot10^4$ | 0.997 | 0.692 | $-1.9\cdot10^3$ | 573.4 | 0.98 | 0.98 | Coll. |
| | $10^{-3}$ | 1 | 0.502 | 0.578 | 0.462 | 0.464 | -0.148 | 0.737 | 0.04 | 0 | No Sig. |
| | $10^{-4}$ | 50 | -0.184 | -0.034 | **0.665** | 0.456 | -0.161 | 0.724 | 0.02 | 0 | **Mem.** |
| bikewale | $10^{-1}$ | 1 | $2.8\cdot10^4$ | $2.2\cdot10^4$ | 0.374 | 0.481 | 20.47 | 78.79 | 0.02 | 0 | Coll. |
| | $10^{-2}$ | 50 | $7.4\cdot10^4$ | $5.8\cdot10^4$ | 0.350 | 0.597 | 55.54 | 168.7 | 0.02 | 0 | Coll. |
| | $10^{-3}$ | 1 | 0.734 | 0.959 | 0.518 | 0.486 | -0.039 | 0.333 | 0.08 | 0.04 | Coll. |
| | $10^{-4}$ | 50 | -0.052 | 0.009 | **0.578** | **0.563** | -0.017 | 0.156 | 0 | 0 | **Mem.** |
| chocolate_bar_ratings | $10^{-1}$ | 50 | -0.024 | 0.165 | **0.585** | **0.594** | -0.007 | 0.063 | 0.16 | 0 | **Mem.** |
| | $10^{-2}$ | 50 | -0.010 | 0.145 | **0.604** | **0.605** | 0.004 | 0.041 | 0.16 | 0 | **Mem.** |
| | $10^{-3}$ | 10 | 0.021 | 0.162 | **0.609** | **0.606** | 0.009 | 0.085 | 0.18 | 0.02 | **Mem.** |
| | $10^{-4}$ | 10 | 0.099 | 0.039 | 0.466 | **0.589** | 0.054 | 0.310 | 0.10 | 0.02 | **Mem.** |
| buy_buy_baby | $10^{-1}$ | 1 | $9.2\cdot10^3$ | $5.9\cdot10^3$ | 0.998 | 1.000 | -761.1 | 313.9 | 1.00 | 0.96 | Coll. |
| | $10^{-2}$ | 10 | $3.8\cdot10^4$ | $2.4\cdot10^4$ | 1.000 | 1.000 | $-2.8\cdot10^3$ | 722.2 | 1.00 | 1.00 | Coll. |
| | $10^{-3}$ | 1 | 0.452 | 0.269 | **0.553** | **0.563** | -0.307 | 1.871 | 0.10 | 0 | **Mem.** |
| | $10^{-4}$ | 10 | -0.060 | -0.011 | **0.608** | **0.577** | -0.213 | 1.791 | 0.14 | 0.12 | **Mem.** |
| cardekho | $10^{-1}$ | 1 | 356.1 | 537.5 | 0.351 | 0.041 | 0.902 | 2.084 | 0.04 | 0 | Coll. |
| | $10^{-2}$ | 1 | 7.705 | 12.75 | 0.546 | 0.540 | -0.026 | 0.324 | 0.24 | 0.06 | Coll. |
| | $10^{-3}$ | 1 | 0.808 | 0.864 | 0.558 | 0.556 | 0.009 | 0.070 | 0.24 | 0.06 | Coll. |
| | $10^{-4}$ | 1 | 0.010 | 0.028 | **0.596** | 0.497 | 0.011 | 0.067 | 0.14 | 0 | **Mem.** |
| beer_ratings | $10^{-1}$ | 1 | $3.3\cdot10^4$ | $2.8\cdot10^4$ | 0.019 | 0.294 | 286.4 | 119.2 | 0 | 0 | Coll. |
| | $10^{-2}$ | 1 | 5.249 | 4.765 | 0.480 | 0.511 | -0.003 | 1.558 | 0.08 | 0.04 | Coll. |
| | $10^{-3}$ | 1 | 1.462 | 1.265 | 0.481 | 0.495 | -0.007 | 0.460 | 0.10 | 0 | No Sig. |
| | $10^{-4}$ | 10 | -0.198 | 0.121 | 0.507 | 0.509 | -0.039 | 0.530 | 0.08 | 0.02 | No Sig. |
| bikedekho | $10^{-1}$ | 1 | $3.7\cdot10^3$ | $3.0\cdot10^3$ | — | — | 57.83 | 16.07 | 0 | 0 | Coll. |
| | $10^{-2}$ | 1 | 2.986 | 1.705 | 0.428 | 0.436 | 0.057 | 0.361 | 0.08 | 0 | Coll. |
| | $10^{-3}$ | 1 | 4.262 | 3.423 | 0.537 | 0.532 | -0.035 | 0.236 | 0.10 | 0.06 | Ctx.Ov. |
| | $10^{-4}$ | 10 | -0.005 | 0.018 | 0.542 | 0.530 | -0.005 | 0.117 | 0.14 | 0.06 | Ctx.Ov. |
| coffee_ratings | $10^{-1}$ | 10 | 9.375 | 9.335 | 0.498 | 0.519 | 0.113 | 2.717 | 0.08 | 0 | Coll. |
| | $10^{-2}$ | 50 | 0.307 | 0.267 | 0.528 | 0.536 | 0.004 | 0.106 | 0.08 | 0.02 | Inv. |
| | $10^{-3}$ | 10 | 0.177 | 0.137 | **0.560** | 0.508 | -0.006 | 0.153 | 0.20 | 0 | **Mem.** |
| | $10^{-4}$ | 50 | 0.182 | 0.292 | 0.520 | 0.529 | 0.011 | 0.173 | 0.08 | 0.02 | Inv. |

# D. Memorization under Realistic Training Regimes

**Memorization Signal under Realistic Regimes** ($Q$=512). Tab. 12 shows how memorization degrades as the training regime progressively approaches practical conditions. Under single-task fine-tuning on fixed context-query pairs (worst case), 5 of 10 tasks are detectable. By fine-tuning a single model jointly on all 10 tasks with fixed context-query pairs,

*Table 11.* Exhaustive evaluation of 8 CARTE datasets under ICLMEM with $Q = 512$ context queries. For each dataset, we track metrics across all four learning rates, selecting the peak-signal epoch independently for $Q = 512$. $\Delta\mathcal{L}_v$: vanilla loss delta $\mathcal{L}_v(M_\theta) - \mathcal{L}_v(M_{ref})$ for members (IN) and non-members (OUT). **N**: normal label condition; **S**: member-shuffled condition. AUC($\Delta S$) and TPR are computed on instance-level delta scores $\Delta S(q) = S(q) - S_{ref}(q)$; bold AUC = detectable memorization (AUC($\Delta S$) > 0.5). **Pert.**: mean $\mu_{gap}$ and std. $\sigma_{gap}$ of the per-manipulation member–non-member probe loss gap $g_k$ across all $\pi_k \in \Pi$. **Outcome**: **Mem.** = Detectable Memorization, Coll. = Collapse, Inv. = Inversion, Ctx.Ov. = Context Overfitting, No Sig. = No Signal.

| Dataset | LR | Ep. | $\Delta\mathcal{L}_v$ IN | OUT | AUC($\Delta S$) N | S | Pert. $\mu_{gap}$ | $\sigma_{gap}$ | TPR @10% | @1% | Outcome |
|---|---|---|---|---|---|---|---|---|---|---|---|
| babies_r_us | $10^{-1}$ | 1 | 206.1 | 171.3 | 0.276 | — | 18.09 | 31.07 | 0.02 | 0 | Coll. |
| | $10^{-2}$ | 1 | 0.711 | 0.742 | 0.487 | 0.463 | -0.075 | 1.234 | 0.12 | 0.01 | Coll. |
| | $10^{-3}$ | 1 | 0.569 | 0.535 | **0.668** | — | -0.102 | 1.285 | 0.21 | 0.01 | **Mem.** |
| | $10^{-4}$ | 10 | -0.033 | 0.004 | **0.608** | 0.482 | -0.085 | 1.289 | 0.02 | 0 | **Mem.** |
| bikewale | $10^{-1}$ | 1 | $3.0 \cdot 10^4$ | $2.7 \cdot 10^4$ | 0.193 | 0.121 | 69.14 | 83.08 | 0.00 | 0 | Coll. |
| | $10^{-2}$ | 1 | 49.04 | 42.88 | 0.485 | 0.466 | 0.084 | 2.658 | 0.09 | 0.01 | Coll. |
| | $10^{-3}$ | 1 | 0.408 | 0.412 | **0.582** | — | -0.009 | 0.264 | 0.27 | 0.12 | **Mem.** |
| | $10^{-4}$ | 10 | -0.021 | 0.008 | 0.536 | 0.521 | -0.009 | 0.264 | 0.11 | 0.01 | No Sig. |
| chocolate_bar_ratings | $10^{-1}$ | 50 | 2.677 | 1.956 | 0.496 | 0.511 | 0.008 | 1.122 | 0.09 | 0.01 | No Sig. |
| | $10^{-2}$ | 50 | 0.067 | 0.025 | 0.514 | 0.516 | 0.000 | 0.056 | 0.10 | 0.02 | Inv. |
| | $10^{-3}$ | 10 | 0.092 | 0.200 | 0.512 | — | -0.001 | 0.175 | 0.10 | 0.02 | Ctx.Ov. |
| | $10^{-4}$ | 10 | 0.009 | -0.012 | 0.516 | 0.510 | -0.000 | 0.055 | 0.09 | 0.01 | Ctx.Ov. |
| buy_buy_baby | $10^{-1}$ | 1 | $1.5 \cdot 10^4$ | $1.7 \cdot 10^4$ | — | — | $2.5 \cdot 10^3$ | 686.5 | 0 | 0 | Coll. |
| | $10^{-2}$ | 1 | 19.78 | 27.81 | 0.431 | 0.424 | 6.079 | 35.42 | 0.05 | 0 | Coll. |
| | $10^{-3}$ | 50 | 0.764 | 0.741 | 0.513 | — | -0.069 | 2.860 | 0.14 | 0.03 | Coll. |
| | $10^{-4}$ | 50 | -0.024 | -0.002 | 0.461 | 0.521 | -0.002 | 2.326 | 0.11 | 0.02 | No Sig. |
| cardekho | $10^{-1}$ | 10 | $5.7 \cdot 10^7$ | $6.0 \cdot 10^7$ | 0.464 | 0.536 | $1.2 \cdot 10^3$ | $1.5 \cdot 10^4$ | 0.08 | 0.02 | Coll. |
| | $10^{-2}$ | 10 | $2.0 \cdot 10^3$ | $2.1 \cdot 10^3$ | 0.552 | 0.492 | -0.455 | 4.224 | 0.11 | 0.02 | Coll. |
| | $10^{-3}$ | 50 | 250.3 | 261.8 | 0.489 | — | 0.021 | 1.473 | 0.10 | 0.02 | Coll. |
| | $10^{-4}$ | 50 | 0.043 | 0.056 | **0.560** | 0.489 | 0.001 | 0.066 | 0.03 | 0.00 | **Mem.** |
| beer_ratings | $10^{-1}$ | 10 | $2.3 \cdot 10^7$ | $2.2 \cdot 10^7$ | 0.999 | 0.909 | $-1.9 \cdot 10^5$ | $5.1 \cdot 10^4$ | 1.00 | 1.00 | Coll. |
| | $10^{-2}$ | 50 | $2.6 \cdot 10^3$ | $2.5 \cdot 10^3$ | 0.792 | 0.996 | -19.68 | 26.75 | 0.24 | 0.04 | Coll. |
| | $10^{-3}$ | 1 | 0.492 | 0.470 | 0.502 | — | -0.034 | 0.619 | 0.08 | 0.01 | No Sig. |
| | $10^{-4}$ | 50 | -0.009 | 0.026 | 0.497 | **0.586** | -0.033 | 0.578 | 0.49 | 0.17 | **Mem.** |
| bikedekho | $10^{-1}$ | 1 | $1.0 \cdot 10^4$ | $9.9 \cdot 10^3$ | — | — | 218.2 | 32.48 | 0 | 0 | Coll. |
| | $10^{-2}$ | 1 | 5.159 | 4.631 | 0.442 | 0.473 | 0.101 | 0.643 | 0.07 | 0.01 | Coll. |
| | $10^{-3}$ | 1 | 1.029 | 1.036 | 0.473 | — | 0.013 | 0.228 | 0.13 | 0.02 | Coll. |
| | $10^{-4}$ | 10 | -0.025 | 0.007 | **0.579** | 0.497 | 0.006 | 0.190 | 0.15 | 0.02 | **Mem.** |
| coffee_ratings | $10^{-1}$ | 50 | 0.284 | 0.246 | 0.519 | 0.517 | -0.004 | 0.156 | 0.09 | 0.00 | Ctx.Ov. |
| | $10^{-2}$ | 50 | 0.263 | 0.223 | 0.519 | 0.522 | -0.003 | 0.142 | 0.09 | 0.00 | Ctx.Ov. |
| | $10^{-3}$ | 10 | 0.178 | 0.134 | 0.521 | — | -0.002 | 0.074 | 0.10 | 0.00 | Ctx.Ov. |
| | $10^{-4}$ | 50 | 0.275 | 0.128 | 0.522 | 0.525 | -0.001 | 0.079 | 0.10 | 0.01 | Ctx.Ov. |

i.e., 10 gradient steps per epoch, the signal appears only for 2 tasks (babies_r_us: AUC 0.668→0.618; bikedekho: AUC 0.579→0.558). We intentionally fixed the context-query pairs across epochs, however, in practice the pairs are randomly sampled. With random context-query sampling and fine-tuning on all 10 tasks, the signal collapses to a single task: bikewale (AUC 0.575). Critically, this residual signal appears only at Epoch 10, already beyond the 2–5 epoch budget of real LTMs pre-training (Spinaci et al.); within that practical budget, no task would be detectable. The pattern confirms that memorization under ICLMEM requires two structural conditions absent from practical pre-training: (i) per-task gradient specialization and (ii) repeated exposure to fixed context-query pairs.

*Table 12.* Peak detectable memorization at $Q{=}512$, for datasets with signal in at least one configuration. *Single-Task Fixed Pairs*: one model fine-tuned per dataset (peak across all $\eta$; $^{\star}\eta = 10^{-3}$, others $\eta = 10^{-4}$). *Multi-Task Fixed Pairs*: one model trained jointly on all 10 CARTE tasks with fixed context-query pairs ($\eta = 10^{-4}$). *Multi-Task Random Pairs*: same joint setup with a fresh random context-query sample at each step ($\eta = 10^{-4}$). **Bold**: highest AUC per row.

| Dataset | Single-Task Fixed Pairs | | | | Multi-Task Fixed Pairs | | | | Multi-Task Random Pairs | | | |
|---|---|---|---|---|---|---|---|---|---|---|---|---|
| | Ep | AUC | TPR$_{@10\%}$ | TPR$_{@1\%}$ | Ep | AUC | TPR$_{@10\%}$ | TPR$_{@1\%}$ | Ep | AUC | TPR$_{@10\%}$ | TPR$_{@1\%}$ |
| babies_r_us | 1 | $^{\star}$**0.668** | 0.205 | 0.008 | 100 | 0.618 | 0.008 | 0.000 | | | — | |
| beer_ratings | 50 | **0.586** | 0.494 | 0.166 | | — | | | | | — | |
| bikedekho | 10 | **0.579** | 0.148 | 0.016 | 1 | 0.558 | 0.190 | 0.027 | | | — | |
| bikewale | 1 | $^{\star}$**0.582** | 0.266 | 0.123 | | — | | | 10 | 0.575 | 0.100 | 0.018 |
| cardekho | 50 | **0.560** | 0.029 | 0.002 | | — | | | | | — | |

