# OpenReview forum: "Probing Memorization of Tabular In-Context Learning"
_ICML.cc/2026/Workshop/FMSD — FMSD @ ICML 2026 Poster_

### Official Review · Reviewer_1dHB · 2026-05-20
**Evaluation of memorization risks in tabular foundation models.**

**Rating:** 5
**Confidence:** 4

**Review:**

### Summary
The authors present ICLMEM, a method for probing memorization in tabular in-context learning. Using their method, it is shown that the TFM ConTextTab can be probed for memorization in specific downstream settings.

### Strengths
The motivation of the work is thorough with various distinctions between memorization in LLMs and tabular models being actively considered. Additionally distinctions are not only highlighted, but reasonable measure are taken to counter problems specific to tabular models, like the ideas about how to pre-process the data to avoid misinterpreting memorization signals.

I appreciate the explicit consideration of difficulty calibration of a task which can be an easy confounder for misidentified memorization otherwise.

Further the detailed study on the drivers of memorization, specifically evaluating the relation of target cardinality and also evaluating the emergence of different properties over a range of fine-tuning epochs is quite insightful.

### Areas for Improvement
One of my main issues with this work is that at times, the ordering and flow can be quite confusing. While I appreciate the detailed consideration of many different caveats like how memorization in LLMs differ to tabular models, how we explicitly need to construct the Context in order to allow to test for memorization, as mentioned before, sometimes the narrative itself is quite hard to follow.

Regarding the robustness measure: while I think it is important to test for robustness across multiple runs, I am wondering whether the 48 context manipulation can be an additional confounder here. That is, potentially altering strategies can make the task itself easier (or harder) again, leading to false indication of memorization.

For the finetuning step, though it is important to test different setups, some of the choices here seem quite confusing. The result of collapsing representations, when fine-tuning the model on comparably large learning rates $(10^{-1}, 10^{-2})$ specifically in combination with small query sizes seems trivial. The fact that a fine-tuning with a learning rate this high, destroys model capabilities seems unsurprising.

Finally, while it is explicitly mentioned as a limitation, that the evaluation scope is limited as of right now, a comparison on different models would be very much necessary to understand how different models memorize data but more importantly to also understand how strong this potential issue of memorization is in frontier models like TabPFN or TabICL. Adding this would be an important addition to this work.

### Detailed Comments
Please refer to the areas of improvement section. Mainly, the general structure of the work could be clearer, many specific design choices are not entirely clear. Also a comparison on more SOTA models /and potentially a more diverse set of datasets would help to better understand the impact of memorization in tabular models.

### Score
The work itself is interesting and fitting the topic of the workshop. However I would argue that parts of the main results are unclear and specifically, important extensions of the study to state of the art models like TabPFN or TabICL are missing. I will therefore argue for a score of 5. I hope that the comments help the authors to improve their work.

---

### Official Review · Reviewer_C6do · 2026-05-21
**Review of Probing Memorization of Tabular In-Context Learning**

**Rating:** 7
**Confidence:** 4

**Review:**

**Summary:**

The paper studies memorization risks in large tabular models (LTMs), specifically tabular foundation models that perform in-context learning. The authors argue that memorization in LTMs differs from LLM memorization because LTMs are constrained to predict labels from the provided context rather than generate arbitrary text. To study this setting, the paper introduces ICLMEM, a probing framework that attempts to separate context-based prediction from parametric memorization. The core idea is to construct a zero-information multiple-choice context where the query features are repeated with every possible candidate label, forcing the model to rely less on contextual patterns and more on any memorized feature-label associations. The authors evaluate ICLMEM on ConTextTab across 10 CARTE tasks and find moderate memorization signals in controlled fine-tuning settings, especially with small query size, fixed context-query pairs, many epochs, and low-cardinality/binary tasks. However, the memorization signal largely disappears under more realistic multi-task training and random context-query sampling.

**Strengths:**
1. **Strong Workshop Relevance:** The paper directly studies privacy and memorization in tabular foundation models, which is highly relevant to the workshop's themes.
2. **Well-motivated problem:** Memorization in LTMs is an important and timely topic, especially as tabular foundation models increasingly incorporate real-world tabular corpora that may contain sensitive enterprise, healthcare, or financial data.
3. **Clear, Solid Experimental Setting:** ICLMEM is thoughtfully designed around the constraints of tabular ICL models. The zero-information multiple-choice protocol is a clever way to reduce context-based inference and probe for parametric memorization. The paper explicitly addresses common membership-inference pitfalls, including distribution shift, feature contamination, base-rate fallacy, sample difficulty, and unverified membership ground truth.
4. **Useful empirical analysis:** The finding that memorization is strongest under artificial or worst-case fine-tuning conditions, but largely disappears under realistic multi-task/randomized training, is valuable. It suggests concrete ways to reduce memorization risk, such as avoiding fixed context-query pairs, small query sizes, excessive epochs, and single-task over-specialization.

**Areas for Improvement:**
1. **Limited model coverage:** The evaluation focuses on one LTM, ConTextTab. Since memorization behavior may vary significantly across tabular foundation model architectures and training objectives, the paper would be stronger with additional models such as TabPFN-style or TabICL-style models.
2. **Moderate signal strength:** The detected memorization is real but not very strong. The reported AUC values are moderate, and strong TPR at low FPR appears only for a subset of tasks/configurations. This should temper the overall privacy-risk claim. This also further motivates the case to increase model coverage - which may provide additional dimensions (eg: model size) to study the impact of memorization.
3. **Attack realism needs clarification:** The zero-information multiple-choice context is methodologically useful, but it is somewhat artificial. A real attacker may not know the candidate label set or be able to construct such contexts, especially for regression or high-cardinality targets. More effort can be made to identify how "real" attacks look in the context of these models.
4. Regression handling could be clearer: The paper applies ICLMEM to both classification and regression tasks, but the multiple-choice framing is more natural for classification. For regression/high-cardinality targets, the paper could more explicitly explain how the candidate label set is constructed and how this affects attack realism and interpretation.

**Justification of Score:**

Overall, this is a strong and relevant workshop submission. The problem is important, the proposed probing framework is thoughtful, and the experimental setting is solid. The main contribution is a principled way to probe memorization in tabular in-context models and identifies the training regimes where memorization is most likely to appear. The main limitations is that the evaluation is limited to one LTM and 10 tasks, and the strongest signals occur in artificial fine-tuning setups rather than realistic training regimes. However, the paper is transparent about these limitations and provides useful insights.

---

### Official Review · Reviewer_MkWy · 2026-05-21

**Rating:** 7
**Confidence:** 4

**Review:**

**Summary**

The paper studies memorisation in in-context learning based tabular foundation models. The setup studies how much parametric memorization affects model prediction (when the model is finetuned on a particular set of examples). The results point to a mild degree of parametric memorization

**Strength**

The paper tackles an interesting question and approaches analysis of tabular foundation model predictions from a new angle.
The methodology looks sound and the results are new and interesting to the community in my opinion

**Areas for Improvement**

There are two broad directions expanding on which I think the project could benefit in the future.

1. Connections to other studies of foundation models that aim to define the mechanisms of the context-based predictions. This study tries to isolate the parametric part of the model in making the prediction, some prior work in contrast, tried to attribute the context-based prediction. It would be nice to have the two explanations conneted in the future.

2. Extending the finetuning setup above the small-scale existing one, increasing the dataset coverage and size, and the amount of training epochs, would memorization affect TFMs finetuned on datasets with 10k samples in a regular regime (e.g. [20-200) epochs)

**Detailed Comments**

Here are some more concrete suggestions/questions on the two points I've outlined in "Areas for Improvement" section.

- It would be interesting to see connections of the analysis methodology used in this paper with the ones from https://arxiv.org/pdf/2605.06510 or https://arxiv.org/abs/2506.08982 (first indicates that current models often under-utilize layers, second looks at how the attention distribution changes after finetuning). Overall, I have a feeling that these papers, and the present one point towards something like "TFMs mostly rely on context" for prediction -- I think expanding on this may be fruitful
- The scope of finetuning on a single dataset could also be widened: e.g. by studying larger finetuning datasets with more epochs (at least tabarena scale datasets, maybe a bit larger -- its interesting how memorization behaves there, collapse may not be such a prominent problem in scenarious with more data available)

Overall I think this paper provides valuable insights and brings new methodology for analyzing tabular FMs.